# Comparing sources of mobility for modelling the epidemic spread of Zika virus in Colombia

**Daniela Perrotta** [1] *, **Enrique Frias-Martinez**[2], **Ana Pastore y Piontti**[3], **Qian Zhang**[3], **Miguel Luengo-Oroz** [4], **Daniela Paolotti**[5], **Michele Tizzoni** [5], **Alessandro Vespignani** [3]

**1** Laboratory of Digital and Computational Demography, Max Planck Institute for Demographic Research, Rostock, Germany, **2** Universidad Camilo Jose Cela, CAILab, Madrid, Spain, **3** Laboratory for the Modeling of Biological and Socio-technical Systems, Northeastern University, Boston, Massachusetts, United States of America, **4** United Nations Global Pulse, New York, State of New York, United States of America, **5** ISI Foundation, Turin, Italy

\* perrotta@demogr.mpg.de

## Abstract

Timely, accurate, and comparative data on human mobility is of paramount importance for epidemic preparedness and response, but generally not available or easily accessible. Mobile phone metadata, typically in the form of Call Detail Records (CDRs), represents a powerful source of information on human movements at an unprecedented scale. In this work, we investigate the potential benefits of harnessing aggregated CDR-derived mobility to predict the 2015-2016 Zika virus (ZIKV) outbreak in Colombia, when compared to other traditional data sources. To simulate the spread of ZIKV at sub-national level in Colombia, we employ a stochastic metapopulation epidemic model for vector-borne diseases. Our model integrates detailed data on the key drivers of ZIKV spread, including the spatial heterogeneity of the mosquito abundance, and the exposure of the population to the virus due to environmental and socio-economic factors. Given the same modelling settings (i.e. initial conditions and epidemiological parameters), we perform in-silico simulations for each mobility network and assess their ability in reproducing the local outbreak as reported by the official surveillance data. We assess the performance of our epidemic modelling approach in capturing the ZIKV outbreak both nationally and sub-nationally. Our model estimates are strongly correlated with the surveillance data at the country level (Pearson's $r = 0.92$ for the CDR-informed network). Moreover, we found strong performance of the model estimates generated by the CDR-informed mobility networks in reproducing the local outbreak observed at the sub-national level. Compared to the CDR-informed networks, the performance of the other mobility networks is either comparatively similar or substantially lower, with no added value in predicting the local epidemic. This suggests that mobile phone data captures a better picture of human mobility patterns. This work contributes to the ongoing discussion on the value of aggregated mobility estimates from CDRs data that, with appropriate data protection and privacy safeguards, can be used for social impact applications and humanitarian action.

**Data Availability Statement:** The mobile phone dataset consists of the weekly origin-destination matrices of the number of trips between municipalities in Colombia and the weekly number

of active phone numbers in each municipality, over a six-month period, from December 2, 2013 to May 19, 2014. The dataset has been used in the manuscript to generate the CDR-informed network and to calibrate the hybrid radiation network. The dataset is proprietary to Telefónica Investigación y Desarrollo in Madrid (Spain). The dataset is subject to strict privacy regulations. Access was granted after signing a non-disclosure agreement (NDA) with the proprietor, who anonymized and aggregated the original data before giving access to the authors. The mobile phone data could be available on request after a NDA is signed and discussed. Contact details for accessing the data is on https://www.telefonica.com/en/sustainability-innovation/innovation/telefonica-research/.

**Funding:** A.V. is partially funded by NIH/NIGMS under grant R01GM130668-01. The findings and conclusions in this study are those of the authors and do not necessarily represent the official position of the funding agencies, the National Institutes of Health or the US Department of Health and Human Services. The funders had no role in study design, data collection and analysis, decision to publish, or preparation of the manuscript.

**Competing interests:** The authors have declared that no competing interests exist.

## Author summary

Human mobility plays a key role in the spread of many infectious diseases. Integrating this variable into spatial epidemic models can provide valuable insights for epidemic preparedness and response. Yet, there are numerous limitations and pitfalls often driven by data scarcity, especially in developing countries. To improve our understanding of the potential benefits of different human mobility data for outbreak prediction, in this work we focused on the aggregated mobility patterns derived from Call Detail Records (CDRs) data in comparison to more traditional data, including census data and mathematical mobility models. Using the 2015–2016 Zika virus (ZIKV) outbreak in Colombia as a case study, we employed a stochastic metapopulation model for vector-borne disease to simulate the ZIKV spread at the sub-national level in Colombia and assess the performance of each mobility network in capturing the ZIKV outbreak both nationally and sub-nationally. We found evidence that the population movements derived from aggregated CDRs data better capture the mobility and mixing patterns relevant to predict the local spread of ZIKV infections.

## Introduction

In 2015–2016, a large-scale outbreak of Zika virus (ZIKV) infection affected the Americas and the Pacific. The epidemic was first confirmed in Brazil in May 2015 and rapidly reached a total of 50 countries and territories through the end of 2016 [1]. ZIKV infection is typically accompanied by mild illness, but following the increased incidence of neurological complications, including microcephaly in newborns and Guillain-Barrè syndrome, the WHO declared a Public Health Emergency of International Concern (PHEIC) [2] in February 2016, which lasted for nearly 10 months.

First isolated in the Zika forest of Uganda in 1947, ZIKV is primarily transmitted by infected *Aedes* mosquitoes [3, 4], also responsible for transmitting other infectious diseases, including dengue, chikungunya, and yellow fever. Other ways of transmission have been reported, such as sexual and perinatal transmission [5–8] and blood transmission through blood transfusion [9]. The likelihood of sustained local transmission of ZIKV is therefore fuelled by the presence of *Aedes* mosquitoes, whose spatial heterogeneity and seasonal variability are in turn regulated by the local environment and climate [10]. Since mosquitoes cannot fly too far, but tend to spend their lifetime around where they emerge, human population movement is likely responsible for ZIKV introduction to new regions with favourable local conditions for mosquitoes proliferation and sustained disease transmission [11].

Human mobility is in fact a key driver of ZIKV spread as well as of several other infectious diseases, increasing disease transmission by introducing new pathogens into susceptible populations, or by increasing social contacts between susceptible and infected individuals [12]. Timely, accurate, and comparative data on human mobility is therefore of paramount importance for epidemic preparedness and response, but generally not available or easily accessible. Traditional data, typically collected from censuses, is often inadequate due to lack of spatial and temporal resolution, or may be completely unavailable in developing countries. Mathematical models, such as the gravity model of migration or the radiation model, represent an alternative to overcome scarcity of traditional data by synthetically quantifying mobility patterns at different scale. However, more detailed data on mixing patterns is generally needed to capture the spatio-temporal fluctuations in disease incidence [13, 14].

The recent availability of large amounts of geolocated datasets have revolutionized the research field, enabling to quantitatively study individual and collective mobility patterns as generated by human activities in their daily life [15]. In this context, mobile phone metadata, typically in the form of Call Detail Records (CDRs), represents a powerful source of information on human movements. Created by telecom operators for billing purposes and summarising customers' activity (e.g. phone calls, text messages and data connections), CDRs represents a relatively low-cost resource to draw a high-level picture of human mobility patterns at an unprecedented scale [12]. The availability of aggregated CDR-derived mobility has impacted several research fields [16], with significant applications to the spatial modelling of many infectious diseases, such as malaria [17, 18], dengue [19], cholera [20], rubella [21], Ebola [22, 23], ZIKV [24], and COVID-19 [25–29].

In this study, we investigate the potential benefits of harnessing CDRs data to predict the spatio-temporal spread of Zika virus in Colombia, at sub-national level, during the 2015–2016 outbreak in the Americas [30]. We assess the potential improvement in predictive power of integrating aggregated cell phone-derived population movements into a spatially structured epidemic model, when compared to more traditional methods (e.g. census data and mobility models). For this, we examine different sources of human mobility, including i) CDRs data, derived from more than two billion encrypted and anonymized calls made by around seven million mobile phone users in Colombia over a six-month period between December 2013 and May 2014 [31]; ii) daily commuting patterns from the 2005 Colombian census [32]; iii) the gravity model, which assumes that the number of trips increases with population size and decreases with distances [33]; and iv) the radiation model, which mainly assumes that mobility depends on population density [34]. After examining their ability to match the census patterns from a network's point of view, we examine whether the observed discrepancies between networks affect the epidemic outcomes. To this end, we employ a metapopulation epidemic model to simulate the spatial spread of Zika virus as governed by the transmission dynamics of the virus through human-mosquito interactions and as promoted by population movements across the country. We find that the model estimates generated by the CDR-informed mobility network in reproducing the local outbreak observed at the sub-national level outperform the results generated by using other mobility networks. This evidence indicates that mobile phone data provides a timely and accurate picture of the human mobility patterns needed to inform infectious disease models. The results presented here lay out the additional value provided by CDRs data that, with appropriate data protection and privacy safeguards, has potential impact in modelling approaches and data analysis with policy making relevance.

## Materials and methods

### Epidemiological data

We use weekly epidemiological reports from the Colombian National Institute of Health (INS) [35] that document the cumulative number of laboratory-confirmed and suspected cases of Zika virus disease by departments and districts (i.e. the major cities of Barranquilla, Buenaventura, Cartagena, and Santa Marta). Reports are accessible at the following URL: http://www.ins.gov.co/buscador-eventos/BoletinEpidemiologico/Forms/AllItems.aspx.

From this, we computed the weekly number of new ZIKV cases by department for the entire epidemic period, from the earliest reported cases in epidemiological week 2015–40 to epidemiological week 2016–40 (note that the INS declared the end of the epidemic on July 25, 2016, in week 2016–30). The incidence data reported by district was included in the total number for the corresponding department. Due to the lack of data in the 2015–47 epidemiological report, suspected cases are calculated by interpolation. Note that the INS did not report the

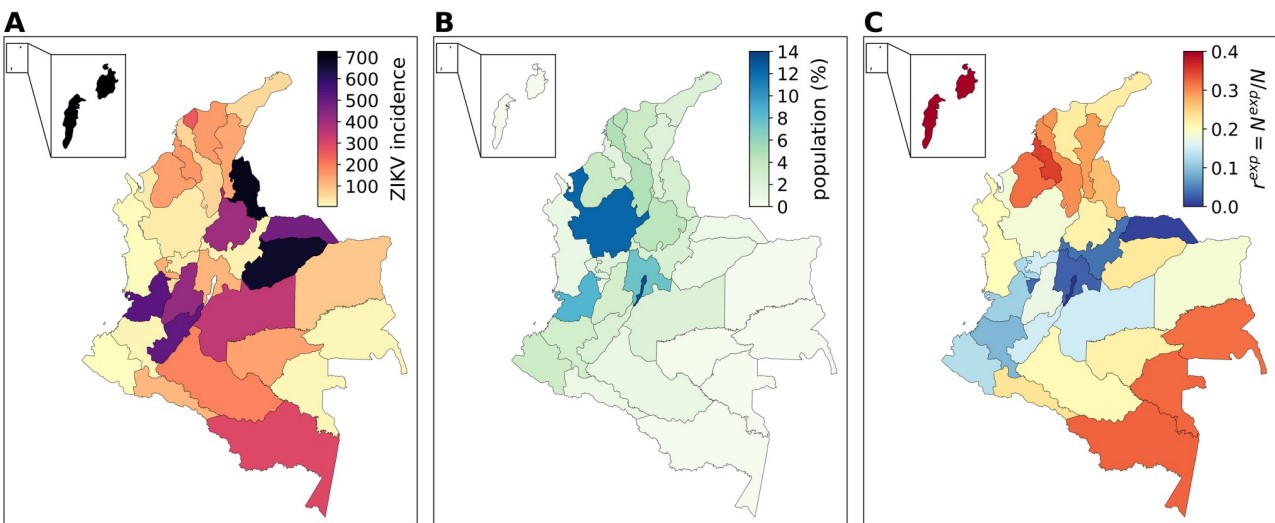

**Fig 1. Data layers by Colombian department.** (A) Cumulative ZIKV incidence (per 100,000 population) reported by Colombia's National Institute of Health in the period from October 4, 2015 (epidemiological week 2015–40) to October 2, 2016 (epidemiological week 2016–40). (B) Population estimates by department. Population is mainly concentrated in the northern and western part of the country, where most of the urban centres are located, whereas the southern and eastern parts of Colombia are mostly sparsely inhabited. (C) Fraction of population exposed to ZIKV due to environmental and socio-economic conditions (more details in Section 3.1 in S1 Appendix).

incidence in the Capital District, Bogotá, since most of the cases in the city originated in other reporting areas.

With over 100,000 cases reported (of which approximately 8% laboratory confirmed), Colombia had the second highest number of reported cases among the 50 countries with autochthonous transmission during the 2015–2016 outbreak in the Americas. Data profiles by department of Colombia are reported in Table A in S1 Appendix. Fig 1A shows the cumulative incidence of Zika virus cases per 100,000 population. The most affected areas were the departments of San Andres (727 cases/100,000), Norte De Santander (692 cases/100,000), and Casanare (670 cases/100,000). Note that underreporting due to the clinical similarities of mild symptoms associated with Zika, limited diagnostic capabilities, medically unattended cases, and asymptomatic infections (ranging from 50% to 80% [36, 37]), may have contributed significantly to underestimating the actual extent of the epidemic.

## Measuring human mobility in Colombia

In this study, we examine different sources of human mobility in Colombia, including the i) CDR-informed mobility, ii) traditional census data, and iii) mathematical mobility models. From this, we create four different mobility networks of daily population movements between the 33 departments of Colombia. Note that, since we use a Markovian dynamics to model the migration process in the epidemic model (more details in the following Section), we symmetrize the flows in each mobility network by averaging flows $w_{ij}$ and $w_{ji}$ (missing links are treated as null values, i.e. $w_{ij} = 0$).

Population data is obtained from the database of the Gridded Population of the World project from the Socioeconomic Data and Application Center at Columbia University (SEDAC), consisting of population estimates in 2015 per grid-cell 1kmx1km (sedac.ciesin.columbia.edu). Fig 1B shows the distribution of population estimates by department.

**CDR-informed mobility network.** We use aggregated mobile phone data obtained from more than two billion encrypted and anonymized metadata calls made over a six-month

 

period, from December 2, 2013 to May 19, 2014. The data consists of weekly origin-destination (OD) matrices of number of trips $T_{ij}^w$ from municipality $i$ to municipality $j$ occurred in week $w$ and weekly number of active phone numbers $n_i^w$ in municipality $i$ in week $w$, where $w$ goes from calendar week 2013–49 to calendar week 2014–21. Note that this data therefore does not refer to daily commuting patterns based on users' most frequently visited locations, but comprise all types of movements. However, given the long observation period and large operator coverage, we assume that potential variability due to long-distance travels, weekly and/or seasonal fluctuations, major vacation periods, etc., are smoothed when considering average values.

From this, we generate the CDR-informed mobility network at the spatial resolution of departments, hereafter referred to as $w_{ij}^{CDR}$, by averaging values over time and normalising flows to match the same population size. In particular, we employ a standard weighting approach and compute weights based on the population sampling ratio $n_i/N_i^w$ in location $i$, where $N_i$ is the resident population (see Fig D in S1 Appendix). This way we correct for potential biases due to under- or over-sampling of the population, although population samples already show good agreement (Spearman's $\rho = 0.87$, $p < 0.01$). More details are provided in Section 2.1 in S1 Appendix.

**Census network.** Commuting data refers to the 2005 Colombian census of the National Institute of Statistics [32]. The data is in the form of an OD matrix of daily population movements between municipalities. We aggregate flows spatially into departments and rescale them to reflect the 2015 population estimates. To this aim, we reweight flows on the population ratio between the 2005 Colombian census and the 2015 population estimates, in order to account for the population changes that occurred over this period. In the following, we will refer to the census network as $w_{ij}^C$. Note that although this dataset is not recent and comprises only the commuting patterns, we will use it as a reference when comparing the various mobility networks.

**Synthetic mobility networks.** We create synthetic mobility networks using two mathematical mobility models, namely the gravity model [33] and the radiation model [34].

The gravity model assumes that the flows $w_{ij}$ of individuals travelling from location $i$ with population $N_i$ to location $j$ with population $N_j$ placed at distance $d_{ij}$ take the following form [33]:

$$w_{ij}^G = C\frac{N_i^\alpha N_j^\gamma}{f(d_{ij})} \tag{1}$$

where $C$ is a proportionality constant, $\alpha$ and $\gamma$ tune the dependence with respect to each location size, and $f(d_{ij})$ is a distance-dependent function. By applying a multivariate linear regression analysis on a logarithmic scale, we estimate the free parameters in Eq (1) that best fit the census data (see Table B in S1 Appendix).

In the radiation model, instead, the flows $w_{ij}$ take the following form [34]:

$$w_{ij}^R = T_i\frac{N_i N_j}{(N_i + s_{ij})(N_i + N_j + s_{ij})} \tag{2}$$

where $N_i$ is the population living at origin $i$, $N_j$ is the population living at destination $j$, $s_{ij}$ is the total population living in a circle of radius $d_{ij}$ centred at $i$, excluding the populations of origin and destination locations, and $T_i$ is the total outflow from $i$ (i.e. $\sum_{j \neq i} w_{ij}$). The radiation model is parameter-free (i.e. it does not require regression analysis or fit on existing data), it only requires the estimate of the total number of travellers $T_i$ from the census data.

 

Given these quantities, we apply the gravity law of Eq (1) and the radiation law of Eq (2) on a fully connected synthetic network, whose nodes correspond to the Colombian departments, thus yielding the flows $w_{ij}^{G}$ and $w_{ij}^{R}$, respectively.

As a sensitivity analysis, since the most recent census data dates back to 2005, we create an alternative radiation network where $T_i$ is calibrated on the CDR-informed mobility data instead of the census data. This requires only highly aggregated information about the total outflows in each department that serve as a rescaling factor to the flows based on the population density. Hence, the goal here is to assess whether very aggregated information obtained from the CDRs, in contrast to the full dataset, may be sufficient to improve the quality of our epidemic modelling estimates. In the following, we will refer to this network as hybrid radiation network and denote it as $w_{ij}^{R,CDR}$. A comparison between the two radiation networks is provided in Section 2.3 in S1 Appendix, while we report the results of the epidemic model in the remainder of the main text. Note that a similar calibration could be used also for the gravity model but without added value in terms of data sharing as we would still need the complete origin-destination matrix from CDRs to fit the model.

## Modelling the epidemic spread of ZIKV in Colombia

We employ a stochastic metapopulation epidemic model to simulate the spatial spread of ZIKV at sub-national level in Colombia as governed by the transmission dynamics through human-mosquito interactions and population movements across the country. In this work we largely follow the state-of-the-art modelling approach of the Global Epidemic and Mobility Model (GLEAM) [38] in the analysis of the 2015–2016 ZIKV epidemic in the Americas developed by Zhang et al. [39]. In this section, we present the conceptual framework while a detailed description is provided in Section 3 in S1 Appendix.

Fig 2A describes the epidemic modelling framework. In the metapopulation structure, the 33 departments of Colombia represent the subpopulations which are coupled by weighted links based on each mobility network considered in this study. The migration process among subpopulations is modelled with a Markovian dynamics, representing individuals who are indistinguishable regarding their travel pattern, so that at each time step the same travelling

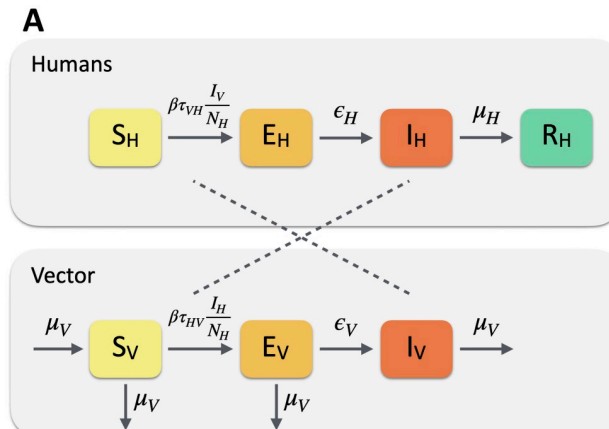

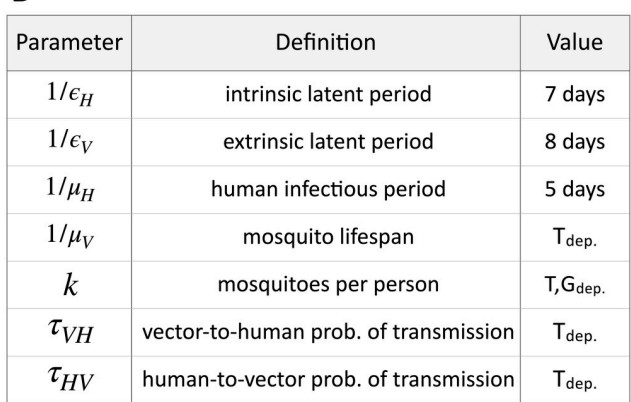

**B**

| Parameter | Definition | Value |
|---|---|---|
| $1/\epsilon_H$ | intrinsic latent period | 7 days |
| $1/\epsilon_V$ | extrinsic latent period | 8 days |
| $1/\mu_H$ | human infectious period | 5 days |
| $1/\mu_V$ | mosquito lifespan | $T_{dep.}$ |
| $k$ | mosquitoes per person | $T,G_{dep.}$ |
| $\tau_{VH}$ | vector-to-human prob. of transmission | $T_{dep.}$ |
| $\tau_{HV}$ | human-to-vector prob. of transmission | $T_{dep.}$ |

**Fig 2. Epidemic modelling framework.** (A) The disease dynamics occurs according to a compartmental classification for ZIKV infection. Humans follow a susceptible-exposed-infectious-removed (SEIR)$_H$ classification, whereas mosquitoes follow a susceptible-exposed-infectious (SEI)$_V$. The transmission dynamics of ZIKV occurs through the interaction between susceptible humans $S_H$ and infected mosquitoes $I_V$, and between infected humans $I_H$ and susceptible mosquitoes $S_V$. (B) Summary of epidemiological parameters: $T_{dep.}$ denotes parameters that are temperature-dependent. $T,G_{dep.}$ denotes parameters that are temperature- and geolocation-dependent. Specific values for the parameters can be found in Refs. [39, 41–43]

probability applies to all individuals without having memory of their origin [40]. No other type of movement is considered. The infection dynamics occurs in homogeneous mixing approximation within each subpopulation according to a compartmental classification of the individuals based on the various stages of the disease. Specifically, humans are classified according to a susceptible-exposed-infectious-removed $(SEIR)_H$ compartmental model, whereas mosquitoes follow a susceptible-exposed-infectious $(SEI)_V$ compartmental model.

The model is fully stochastic and transitions among compartments are simulated through chain binomial processes. The transmission dynamics of the virus occurs through the interaction between i) susceptible humans $S_H$ and infected mosquitoes $I_V$ under the vector-to-human force of infection $\lambda_{VH}$, and ii) infected humans $I_H$ and susceptible mosquitoes $S_V$ under the human-to-vector force of infection $\lambda_{HV}$. We neglect the secondary routes of transmission, e.g. perinatal or blood transmission. The force of infection follows the usual mass-action law, given by the expressions $\lambda_{VH} = \beta\tau_{VH}\frac{I_V}{N_H}$ and $\lambda_{HV} = \beta\tau_{HV}\frac{I_H}{N_H}$, where $\beta$ accounts for the daily mosquito biting rate and the specific transmissibility of ZIKV, and $\tau_{VH}$ and $\tau_{HV}$ correspond to the probability of transmission mosquito-to-human and human-to-mosquito, respectively. The remaining transitions between compartments occur spontaneously. Exposed individuals $E_H$ become infectious at a rate $\epsilon_H$ and infectious individuals $I_H$ recover from the disease at a rate $\mu_H$, inversely proportional to the mean infectious period, $\mu_H^{-1}$. Similarly, exposed mosquitoes $E_V$ become infectious at a rate $\epsilon_V$ and die at a rate $\mu_V$, inversely proportional to the mosquito lifespan $\mu_V^{-1}$. Mosquitoes are re-introduced in the susceptible compartment at the same rate to allow the replenishment of mosquitoes after death.

Our epidemic model integrates detailed data on spatial and seasonal heterogeneity driven by the presence of the vector and the exposure of the population to the vector itself due to socio-economic factors. This is because sustained local transmission of Zika virus is possible only in those areas where the local environment and climate favour the proliferation of mosquitoes [10], but at the same time the socio-economic factors modulate the exposure of the population to the vector itself, even when the environmental conditions are suitable for the transmission of the virus. Fig 2B reports a summary of the epidemiological parameters that intervene in the model, accounting for the key drivers of ZIKV transmission, such as temperature and mosquito abundance. These are also used to identify those areas where ZIKV outbreaks are not possible due to environmental factors. Moreover, GDP per capita estimates are used to model the socio-economic heterogeneity and its impact on the population's risk of exposure to mosquitoes. Population is therefore assigned a rescaling factor $r_{se}$ modulating its exposure to the vector based on local socio-economic conditions. Fig 1C shows the fraction of the population exposed to ZIKV due to environmental and socio-economic conditions. More details are reported in Section 3 in S1 Appendix.

The transmission of ZIKV in the Americas was first confirmed in May 2015 in northeast Brazil, but epidemiological and genetic findings estimated that ZIKV arrived in Brazil much earlier, between October 2013 and April 2014 [44]. After that, ZIKV was likely introduced to Colombia between January and April 2015 [45], that is 6 to 9 months before the ZIKV outbreak was officially declared by the Colombian National Institute of Health in October 2015. Traditional disease monitoring was therefore not sufficient to capture the initial spread of infections in Colombia. In the absence of accurate data on the introduction of Zika virus in Colombia and following the evidence that many ZIKV infections were likely imported into Colombia throughout the epidemic [45], we use the simulation output of the computational model (GLEAM) developed by Zhang et al. [39] as initialization of our epidemic model. Following the approach by Sun et al. [46], we extract the travel-associated ZIKV infections entering Colombia as stochastically simulated by GLEAM. This results in a total of 1,189 simulated

ZIKV epidemics for which we know the time of arrival, the stage of ZIKV infection (exposed or infectious), and the airport of origin and arrival. Fig I in S1 Appendix shows the time-series boxplot of Zika virus imported cases, along with the main countries of origin and departments of destination in Colombia. The daily number of ZIKV introductions has a median value of 10 cases (IQR: 3–21) for a total of 8,671 cases (IQR: 8,315–9,064) imported into Colombia during the entire epidemic period. Note that the same rescaling factor due to environmental and socio-economic conditions applies to the imported ZIKV infections such that the likelihood of seeding an epidemic locally varies depending on whether the subpopulation of destination is at risk or not of ZIKV transmission. This is evident in Fig J in S1 Appendix that shows the average daily ZIKV introductions and its proportion rescaled by the overall exposure to the vector.

We generate 100,000 stochastic realizations using discrete time steps of one full day starting on January 1, 2015. At each iteration, we randomly sample one simulated time-series of ZIKV imported cases among the 1,189 simulations and use it as seeding of our epidemic model. The process is repeated for each mobility network under study, so that, given the same modelling settings (i.e. initial conditions and epidemiological parameters), we can assess their performance in predicting the Zika virus outbreak in Colombia.

Data analysis was performed with Python (version 3.7). The code of the epidemic model was written in object-oriented C++ for computational efficiency and the simulations were performed in parallel on a high-performance computing cluster of 11 cores (146 nodes). All maps were generated by manipulating open-source shapefiles of Colombia using the Geopandas library available in Python. The resulting mobility networks generated by the census data, the gravity model, and the radiation model are reported in S2 Appendix.

## Results

### Comparing sources of human mobility in Colombia

Fig 3 shows the mobility networks in form of origin-destination matrices as obtained from the CDR-informed network (A), the census network (B), the gravity network (C), and the radiation network (D). All networks share the same number of nodes (i.e. Colombian departments), but with significant variations in the number of weighted links and total volume of travellers

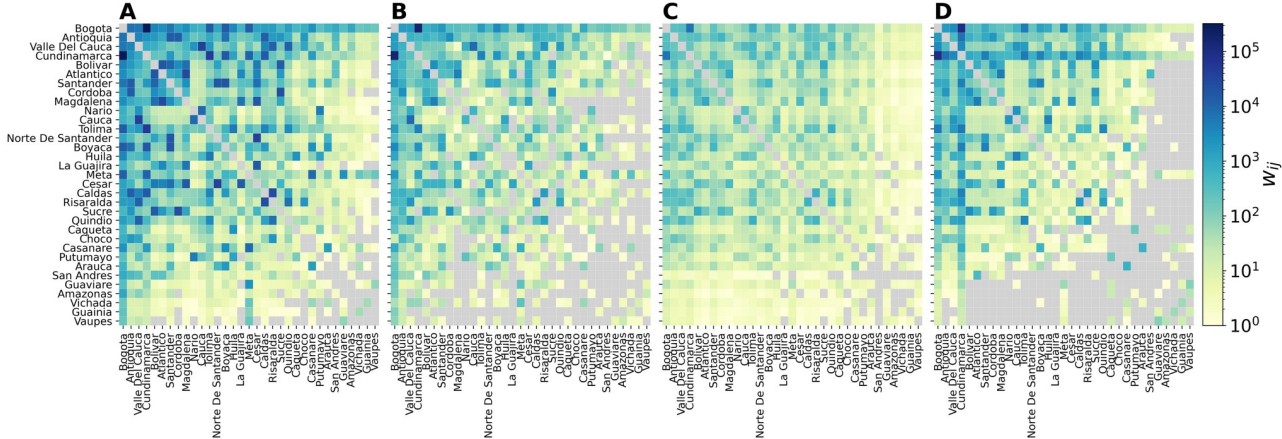

**Fig 3. Mobility networks.** Origin-destination matrices of the flows $w_{ij}$ among Colombian departments in the CDR-informed network (A), the census network (B), the gravity network (C), and the radiation network (D). The colour code represents the weights $w_{ij}$ on links $ij$ (grey indicates no movement). Departments are sorted according to population size.

**Table 1. Basic properties of the mobility networks.** The table reports the total number of nodes and links, the number of links shared with the census network, and the total volume of travellers of each mobility network under study. Self-loops are excluded.

| Network | No. nodes | No. links | No. shared links (%) | Volume |
|---|---|---|---|---|
| $w_{ij}^C$ | 33 | 760 | - | 494,234 |
| $w_{ij}^{CDR}$ | 33 | 972 | 742 (97.63) | 2,005,992 |
| $w_{ij}^G$ | 33 | 1,006 | 754 (99.21) | 71,871 |
| $w_{ij}^R$ | 33 | 736 | 642 (84.47) | 457,737 |

(Table 1). The gravity network has the largest number of links and fully connected nodes, whereas the CDR-informed network has the largest number of travellers. The heatmaps show also that the flows $w_{ij}$ decrease with population size. This is particularly evident in the radiation network (Fig 3D) as the model assumes that mobility depends on population density, thus penalising those departments that are less populated. On the other hand, the gravity network (Fig 3C) is highly connected with smaller flows even between more distant and less populated departments. Mobility flows generally decrease with distance (see Fig C in S1 Appendix). In all networks, the largest flow occurs between the Capital District Bogotá and the near department of Cundinamarca, which is approximately 57 km distant, and concerns most of the commuting pattern. In general, higher mobility rates mainly concern the northern and western part of the country, where most of the urban centres are located, whereas lower rates of mobility concern the southern and eastern parts, which are more sparsely inhabited (see maps in Fig B in S1 Appendix).

Restricting the analysis to the topological intersection of the mobility networks and the census network, we analyse the structural and flows properties of the networks. Table 2 reports the similarity metrics of the mobility networks compared to the census network (definitions are reported in Section 4 in S1 Appendix). Considering the topology of the networks in terms of shared links compared to the total number of links, the Jaccard index is 0.75 for all mobility networks. However, when considering the weights $w_{ij}$, the common part of commuters (CPC) varies significantly across networks, ranging from 0.22 for the gravity network to 0.69 for the radiation network. Finally, the cosine similarity, which is a measure of similarity that takes into account both links and weights shared by two networks, ranges from 0.92 for the gravity network to 0.99 for the radiation network.

Fig 4 shows the mobility flows $w_{ij}$ as compared to the flows $w_{ij}^C$ of the census network. Flows in the CDR-informed network are generally larger than in the census network. Correlation between flows $w_{ij}$ is highest for the CDR-informed network, with Kendall's $\tau = 0.70$ and Spearman's $\rho = 0.88$, while we found weaker correlations for the radiation network ($\tau = 0.58$, $\rho = 0.77$). When considering the outflows $\Sigma_i w_{ij}$, the radiation network shows instead the highest

**Table 2. Statistical comparison of the mobility networks against the census network.** The table reports the values of Kendall's $\tau$ and Spearman's $\rho$ correlation coefficients (computed both on flows $w_{ij}$ and outflows $\Sigma_i w_{ij}$), the Jaccard index, the cosine similarity, and the common part of commuters (CPC). All p-values are statistically significant ($p < 0.01$).

| Network | Kendall $\tau$ | | Spearman's $\rho$ | | Jaccard Index | Cosine Similarity | CPC |
|---|---|---|---|---|---|---|---|
| | $w_{ij}$ | $\Sigma_i w_{ij}$ | $w_{ij}$ | $\Sigma_i w_{ij}$ | | | |
| $w_{ij}^{CDR}$ | 0.70 | 0.77 | 0.88 | 0.92 | 0.75 | 0.97 | 0.39 |
| $w_{ij}^G$ | 0.60 | 0.73 | 0.78 | 0.89 | 0.75 | 0.92 | 0.22 |
| $w_{ij}^R$ | 0.58 | 0.81 | 0.77 | 0.95 | 0.75 | 0.99 | 0.69 |

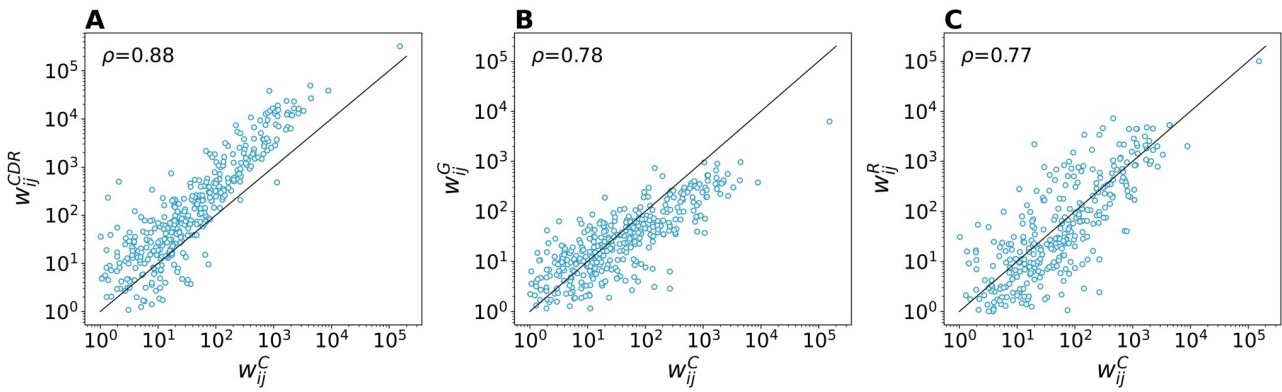

**Fig 4. Comparison of mobility flows against the census network.** Relationship between census flows $w_{ij}^C$ (x-axis) and mobility flows in the CDR-informed network (A), gravity network (B), and radiation network (C). Spearman's $\rho$ correlation coefficient is reported.

correlations ($\tau = 0.81$, $\rho = 0.95$) as the total volume of travellers match the volume in the census network.

## Comparing the mobility networks in the epidemic outcome

Stochastic realisations obtained from our epidemic model (run separately for each mobility network) define the model output we used to describe the spatio-temporal patterns of ZIKV spread in Colombia and assess the potential benefits of using CDR-derived mobility. From this stochastic ensemble, we compute the weekly number of new ZIKV infections (median number and 95% CI) of the model estimates. We first assess the performance of each mobility network in reproducing the outbreak at the national level. In Fig 5, we show the estimated weekly incidence of ZIKV infections (per 100,000 population) in comparison with the official surveillance data reported by Colombia's National Institute of Health (INS). For ease of comparison, the latter is shown on a different y-axis matching the peak of the model estimates of the CDR-informed network. This is because the model projects a much larger number of infections than those captured by the surveillance system, as expected for a typically asymptomatic or mild disease. In particular, based on the official surveillance data, the epidemic peak occurred in week 2016–05 with an incidence of approximately 10 cases per 100,000 population, which results in a case ascertainment rate of about 1% for the reporting system at the peak of the epidemic. To quantify the model performance in capturing the temporal trend of infections, we compute the Pearson's $r$ correlation between the estimated and observed ZIKV incidence at the country level between week 2015–40 and week 2016–40. This ranges between 0.88 for the radiation network to 0.92 for the CDR-informed network (all $p < 0.01$). This is an indicator of the goodness of our model's performance, including its epidemiological assumptions, in capturing the outbreak dynamics without any fit on the observed data. As for the epidemic peak, the model predictions are in good agreement and predict the peak within the confidence intervals. In particular, the model estimates of the radiation network predict the epidemic peak accurately at week 2016–05, with 95%CI ranging from week 2015–51 to week 2016–14. The model estimates of the census and gravity networks predicts the epidemic peak with 1 week lag (2016–06), the CDR-informed network with 4 weeks lags (2016–09), whereas the hybrid radiation network with 5 weeks lags (2016–10).

In order to provide a more detailed analysis of the goodness of fit, among each stochastic ensemble output generated for each mobility network, we select only those stochastic realisations reproducing the observed epidemic peak in Colombia (±1 week). This additional

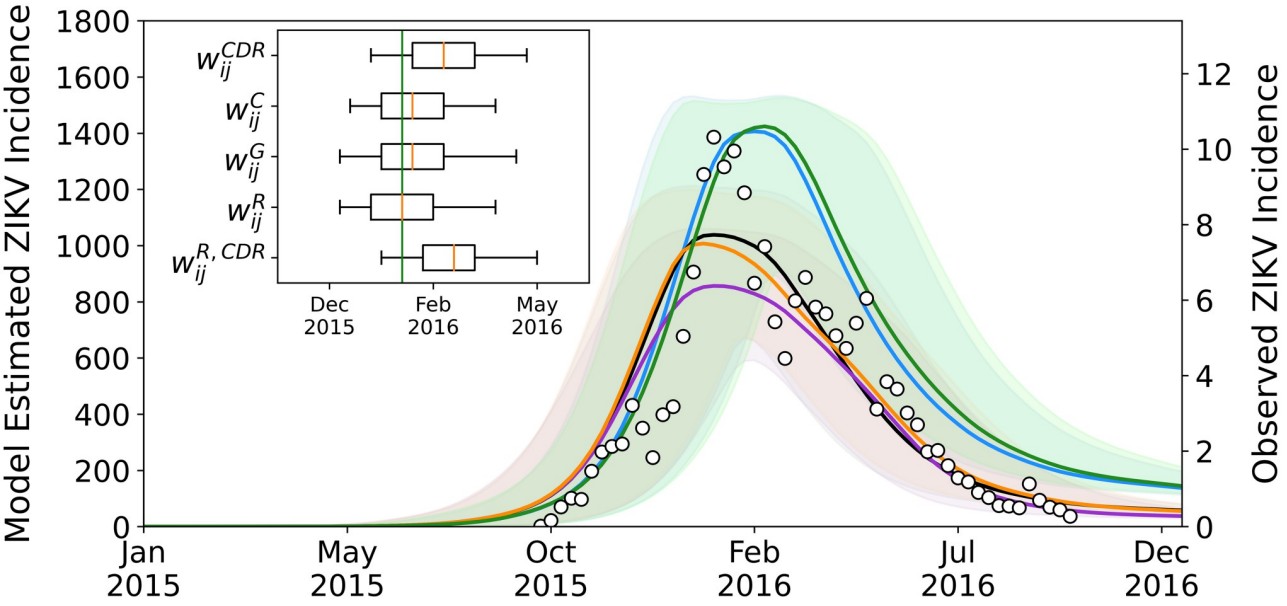

**Fig 5. Comparison between the estimated and observed ZIKV incidence.** Weekly number of new ZIKV infections (per 100,000 population) as estimated from the stochastic ensemble output in the setting using the CDR-informed network (blue), the census network (black), the gravity network (orange), the radiation network (purple), and the radiation network calibrated on the CDR-informed mobility (green). The bold line and shaded area refer to the median number of infections and 95% CI of the model estimates. Black dots correspond to the official ZIKV incidence (per 100,000 population) reported by Colombia's National Institute of Health (right y-axis). For ease of comparison, surveillance data is scaled on the peak of the model estimates of the CDR-informed network. The inset graph shows the peak week as calculated from the model estimates. The observed epidemic peak was in week 2016–05 (green line).

calibration allows us to generate output ensembles with a narrow confidence in the epidemic timing and enables the analysis of simulations at the department level conditional to the occurred national peak timing. Findings are consistent when selecting stochastic realisations with a tolerance of ±2 weeks around the observed epidemic peak.

We excluded from this analysis those departments with less than 100 total ZIKV cases reported by the official surveillance data, which correspond to the departments of Nario, Vichada, Choco, Vaupes, and Guainia (cumulative cases are reported in Table A in S1 Appendix). As for the Capital District Bogotá, ZIKV cases were not reported by the INS since cases mostly originated in other reporting areas, and our model estimates capture this evidence as no new ZIKV infections are generated in this area due to the adverse environmental and socio-economic conditions. This further strengthens our epidemic modelling choices in integrating those factors relevant to reproduce the spread of ZIKV in Colombia. Model estimates are of course affected by the data layers we integrated in our epidemic modelling approach. As expected, model estimates are correlated with the rescaling factor $r^{exp}$ regulating the population exposure to ZIKV due to environmental and socio-economic conditions (Spearman's $\rho$ ranging between 0.69 to 0.73). The model-based projections increase with higher values of $r^{exp}$ as the size of the population participating in the infection dynamics increases (Fig K in S1 Appendix). To compare the total ZIKV cases projected in our model estimates against those observed in the official surveillance data, we estimate a reporting and detection rate through a linear regression fit (see top panel of Fig K in S1 Appendix). The estimated detection rate ranges between 0.51% ± 0.23% for the gravity network to 0.72% ± 0.32% for the CDR-informed network (all $p < 0.05$), thus confirming for the detection and reporting system the ascertainment rate we estimated at the national level.

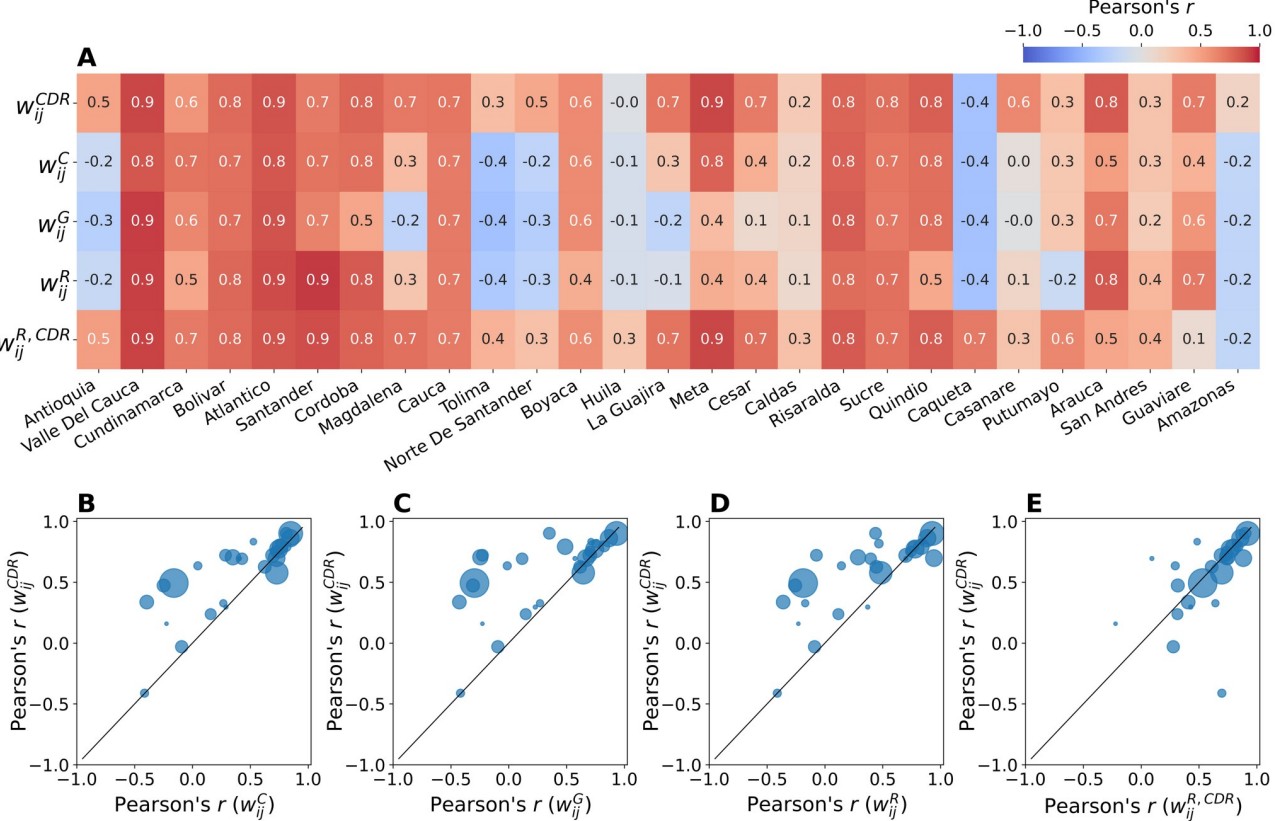

**Fig 6. Correlation between model estimates and official surveillance data.** (**A**) The heatmap shows the Pearson's $r$ correlation obtained by comparing the model estimates generated by each mobility network (on the y-axis) and the official surveillance times series by department (on the x-axis, sorted by population size). The bottom row shows the comparison between the Pearson's $r$ correlation obtained for the CDR-informed network (y-axis) with the Pearson's $r$ correlation obtained for the census network (**B**), the gravity network (**C**), the radiation network (**D**), and the hybrid radiation network (**E**). Point size corresponds to population size.

To quantify the model's performance in capturing the epidemic timing in each Colombian department, we compute the Pearson's $r$ correlation between the model estimates generated by each mobility network and the observed surveillance time series, as shown in Fig 6A. Namely, we investigate the correlation between the model estimated weekly incidence and the corresponding observed surveillance incidence in the time span ranging from week 2015–40 to week 2016–40. The CDR-informed network predicts well the local outbreak in 20 out of 27 departments (i.e. significant correlations), which are all situated in the northern and central part of the country, where most of the population lives. Interestingly, the hybrid radiation network predicts well the local outbreak in 21 out of 27 departments and outperforms the radiation network. The CDR-informed mobility networks thus show similar results, except in the department of Caqueta where all mobility networks fail, but epidemic estimates generated by the hybrid radiation radiation network display higher correlation with the surveillance data (Pearson's $r$ = 0.70). In the remaining departments where the CDR-informed network and the hybrid radiation network fail in reproducing the local outbreak, the other mobility networks do so as well. This is more evident in the bottom row of Fig 6 where we compare the correlation of the CDR-informed network with the correlation of the census network (B), the gravity network (C), the radiation network (D), and the hybrid radiation network (E), by population size. Compared to the CDR-informed network, the performance of the other mobility

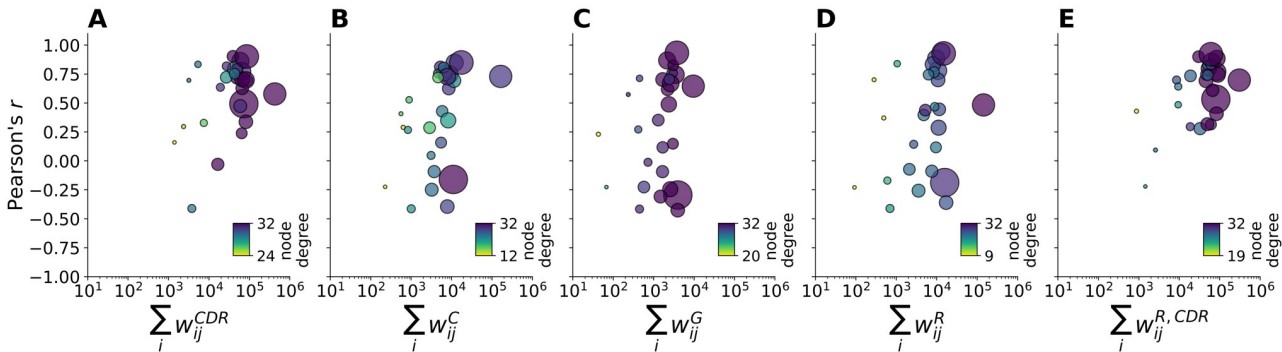

**Fig 7. Correlation by main properties of mobility networks.** The plots show the Pearson's $r$ correlation (y-axis) by the total outflows $\sum_i w_{ij}$ of the CDR-informed network (**A**), census network (**B**), gravity network (**C**), radiation network (**D**), and hybrid radiation network. Point size corresponds to population size. Colour code corresponds to node degree. Note that the scale of the colorbar changes across subplots in order to highlight the variability across networks.

networks is comparatively similar or substantially lower, with no added value in predicting the ZIKV outbreak at the level of departments, with the exception of the hybrid radiation network which shows similar performance.

We investigate this further by looking at the main characteristics of the mobility networks, i.e. node degree, total volume (traffic), and population size, as shown in Fig 7. Here, we observe that in both the CDR-informed network (Fig 7A) and the hybrid radiation network (Fig 7E) correlations are lower in those departments with smaller node degree, lower traffic and smaller population size, which is the case of the departments of Putumayo, Amazonas, and San Andres. On the contrary, the performance of the other mobility networks is very heterogeneous: departments with small values of node degree, traffic and population, reach good results, and vice versa departments with high values perform worse.

## Discussion

To improve our understanding of the potential benefits of different human mobility data for outbreak prediction, in this work we focused on the aggregated CDR-derived mobility data in comparison to more traditional data sources, including census data and mathematical mobility models. Using the 2015–2016 Zika virus (ZIKV) outbreak in Colombia as a case study, we employed a stochastic metapopulation model for vector-borne disease to simulate the ZIKV spread and assess the performance of each mobility network in capturing the ZIKV outbreak both nationally and sub-nationally. Following the state-of-the-art computational modelling approach developed by Zhang et al. [39], our model integrates detailed data on the population, the spatial heterogeneity of the mosquito abundance, and the exposure of the population to the virus due to environmental and socio-economic factors. Moreover, we employed the simulation outputs of the epidemic model by Zhang et al. [39] as initialization of our epidemic model to overcome the lack of official surveillance data in the initial phase of the ZIKV outbreak. This allows us to inform our epidemic model with the travel-associated ZIKV infections entering Colombia and potentially triggering ZIKV transmission depending on the local conditions. Given the same modelling settings (i.e. initial conditions and epidemiological parameters), we performed in-silico simulations for each mobility network and assessed their performance in reproducing the local outbreak as reported by the official surveillance data from the Colombia's National Institute of Health. Our study shows that aggregated information obtained from CDRs data provide the best performance in predicting local outbreaks than the other mobility networks. This suggests that aggregated CDR-informed mobility data better captures the

mobility and mixing patterns relevant to predict the local spread of ZIKV infections. We thoroughly discuss our main results in the following.

First, we showed the performance of our epidemic modelling approach in predicting the ZIKV outbreak at the national level without fitting the model projections on the observed data. Remarkably, we found the model estimates to be strongly correlated with the official surveillance data: the highest correlation is obtained for the CDR-informed network (Pearson's $r = 0.92$), but comparatively similar for the other mobility networks. Moreover, our model estimates do not report ZIKV infections in the Capital District Bogotá, in agreement with the official surveillance data, as the environmental and socio-economic conditions are adverse to local ZIKV spread. This allows us to prove the strength of our epidemic modelling choices in integrating those factors relevant to predicting the ZIKV outbreak in Colombia and to therefore focus on the impact of the human mobility patterns to capture the spatial ZIKV spread, after the simulations' selection.

Second, both mobility networks based on the CDR-informed mobility, i.e. the CDR-informed network and the hybrid radiation network, show the best performance in predicting the epidemic locally. Specifically, the CDR-informed network predicts well the local outbreak in 20 out of 27 departments, whereas the hybrid radiation network predicts well the local outbreak in 21 out of 27 departments and outperforms the radiation network. When the model estimates generated by the CDR-informed networks fail, this is consistent for all mobility networks, as in the case of the departments of Huila and Amazonas. In particular, compared to the CDR-informed networks, the performance of the other mobility networks is either comparatively similar or substantially lower, with no added value in predicting the local epidemic. Specifically, we found that correlations are smaller for the CDR-informed networks in those departments with smaller node degree, lower traffic, and smaller population size. This is the case of the departments of Putumayo, Amazonas, and San Andres. This latter is an archipelago approximately 750 km north of the Colombian mainland, thus having fewer connections and smaller movements with the other departments. On the contrary, the performance of the other mobility networks is very heterogeneous: departments with small values of node degree, traffic and population, show good correlation, and vice versa departments with high values perform worse.

This work comes with several limitations. First, official surveillance data on the ZIKV epidemic suffer from several limitations. Traditional monitoring and reporting of ZIKV infections was not sufficient to capture the introduction of the virus in Colombia. According to genetic findings ZIKV circulated in the Americas since late 2013 [44], but official surveillance began much later in Colombia, in August 2015, months after the epidemic was confirmed in Brazil in May 2015. Moreover, the weekly epidemiological reports from the Colombian National Institute of Health are often inconsistent or inadequate with numbers of cases varying significantly over time and comparatively low detection of laboratory-confirmed cases. Underreporting due to the clinical similarities of mild symptoms associated with ZIKV, limited diagnostic capabilities, medically unattended cases, and asymptomatic infections, may have contributed significantly to underestimating the actual extent of the epidemic. This represents an additional challenge in our study as we use this dataset as a reference to assess the model performance in reproducing the ZIKV outbreak.

Second, the census data employed here refers to the 2005 Colombian census, that is ten years before the Zika virus outbreak in 2015–2016. More recent data may be able to better capture the mobility features of the population and therefore the spatial ZIKV spread. On the other hand, the census data consists of commuting patterns of workers and students who commute daily to their workplace or school. Although this is the official source for trip-level data, this type of mobility is limited to commuting only, typically centred on major urban centres,

and may not be representative of the mobility in rural or distant areas. As an example, in our study the census network performs best in the department of Cundinamarca, which is the nearest department to the Capital District Bogotá. Here the commuting may represent the largest part of the mobility patterns and thus be captured well by census data. In this context, the CDR-informed network may be instead more representative in capturing different types of mobility and not only daily commuting patterns, although inevitably biased by population sampling and coverage. Nonetheless, the census data employed in this work represents the only official data source available at the time of the 2015–2016 Zika virus epidemic in Colombia and the goal of our study is indeed to highlight the limited predictability in epidemic outbreaks in the absence of more refined and updated sources of mobility, such as aggregated CDRs data.

Our modelling approach also contains assumptions and approximations as discussed in Zhang et al. [39]. The transmission model has been calibrated by using data from the French Polynesia outbreak in 2013–2014 and the expressions for temperature dependence of transmissibility are modelled on dengue virus data. Secondary modes of transmission, e.g. perinatal or blood transmission, are not incorporated into the model. Mosquito abundance relies on the mosquito presence/absence maps that come with further limitations [10, 47, 48]. Finally, we do not model public health interventions to control the vector population or behavioural changes due to increased awareness, which we know might be a key aspect in shaping the course of epidemics.

Our findings are in line with a recent study on the 2015–2016 Zika virus epidemic in Colombia showing that an ensemble modelling approach integrating multiple data sources for human mobility, including CDR-derived mobility, is prominent to forecast an emerging infectious disease like Zika [24]. Human mobility is in fact a key driver of ZIKV spread and integrating this variable into spatial models can provide valuable insights for epidemic preparedness and response [11]. Yet, there are numerous limitations and pitfalls often driven by data scarcity, especially in developing countries. Our study shows that even very aggregated information obtained by the CDRs data are sufficient to outperform the epidemic outcomes generated by traditional data sources or mobility models based on such data. In the case of the hybrid radiation network, telecom providers would need to share only highly aggregated information on the total outflows in each department, thus preserving users' privacy. Though the Zika virus outbreak modelled in this work is over in Colombia, in 2022 there are still many countries with autochthonous mosquito-borne transmission—a threat that is increasing due to climate change. The response to many vector-borne diseases could benefit from the proposed modelling approach which should be part of epidemic response toolkits of public health authorities. Furthermore, in the ongoing COVID-19 pandemic, we believe this work is relevant not only because of the proposed methodologies, but also as it contributes to the ongoing discussion on the value of aggregated mobility estimates from CDRs data that, with proper data protection and data privacy mechanisms, can be used for social impact applications and humanitarian action [29].

## Supporting information

**S1 Appendix. Supplementary materials.** Additional descriptions of the methodology and sensitivity analysis on the CDR-derived mobility, the distribution of imported ZIKV cases, and the spatial epidemic modelling approach employed to simulate the spread of ZIKV in Colombia.
(PDF)

**S2 Appendix. Supplementary data.** Resulting mobility networks at the department level in Colombia generated by the census data, the gravity model, and the radiation model.
(ZIP)

## Author Contributions

**Conceptualization:** Daniela Perrotta, Miguel Luengo-Oroz, Michele Tizzoni, Alessandro Vespignani.

**Data curation:** Daniela Perrotta.

**Formal analysis:** Daniela Perrotta.

**Investigation:** Daniela Perrotta, Michele Tizzoni, Alessandro Vespignani.

**Methodology:** Daniela Perrotta, Michele Tizzoni, Alessandro Vespignani.

**Resources:** Enrique Frias-Martinez, Ana Pastore y Piontti, Qian Zhang.

**Software:** Daniela Perrotta.

**Supervision:** Miguel Luengo-Oroz, Daniela Paolotti, Michele Tizzoni, Alessandro Vespignani.

**Validation:** Daniela Perrotta, Michele Tizzoni.

**Visualization:** Daniela Perrotta.

**Writing – original draft:** Daniela Perrotta.

**Writing – review & editing:** Daniela Perrotta, Enrique Frias-Martinez, Ana Pastore y Piontti, Qian Zhang, Miguel Luengo-Oroz, Daniela Paolotti, Michele Tizzoni, Alessandro Vespignani.

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
