## [Decision Letter · Decision Letter 0]

22 Dec 2021

Dear Dr. Perrotta,

Thank you very much for submitting your manuscript "Comparing sources of mobility for modelling the epidemic spread of Zika virus in Colombia" for consideration at PLOS Neglected Tropical Diseases. As with all papers reviewed by the journal, your manuscript was reviewed by members of the editorial board and by several independent reviewers. The reviewers appreciated the attention to an important topic. Based on the reviews, we are likely to accept this manuscript for publication, providing that you modify the manuscript according to the review recommendations. 

This is a very well written manuscript with rigorous analyses. My main comments are for the discussion. The first paragraph could be strengthen with highlight the specifics of the results, what the key findings? And more discussion about how these specific results fit within the literature is needed.

Sincerely,

Kate Zinszer

Associate Editor

Bruce Lee

Deputy Editor

This is a very well written manuscript with rigorous analyses. My main comments are for the discussion. The first paragraph could be strengthen with highlight the specifics of the results, what the key findings? And more discussion about how these specific results fit within the literature is needed.

Reviewer's Responses to Questions

**Key Review Criteria Required for Acceptance?**

**Methods**

-Are the objectives of the study clearly articulated with a clear testable hypothesis stated?

-Is the study design appropriate to address the stated objectives?

-Is the population clearly described and appropriate for the hypothesis being tested?

-Is the sample size sufficient to ensure adequate power to address the hypothesis being tested?

-Were correct statistical analysis used to support conclusions?

-Are there concerns about ethical or regulatory requirements being met?

Reviewer #1: (No Response)

Reviewer #2: (No Response)

**Results**

-Does the analysis presented match the analysis plan?

-Are the results clearly and completely presented?

-Are the figures (Tables, Images) of sufficient quality for clarity?

Reviewer #1: (No Response)

Reviewer #2: (No Response)

**Conclusions**

-Are the conclusions supported by the data presented?

-Are the limitations of analysis clearly described?

-Do the authors discuss how these data can be helpful to advance our understanding of the topic under study?

-Is public health relevance addressed?

Reviewer #1: (No Response)

Reviewer #2: (No Response)

**Editorial and Data Presentation Modifications?**

Reviewer #1: (No Response)

Reviewer #2: (No Response)

**Summary and General Comments**

Reviewer #1: This manuscript tackles two extremely relevant and timely topics at once. On the one hand, the authors present a computational model that is able to reproduce the empirical data on the outbreak of Zika virus in Colombia; most importantly on the other hand, the authors actually address the issue of which kind of mobility data sources are most suited to feed the model. Indeed, while very detailed data sources are becoming available, they are often used somehow "blindly" in models, without the demonstration that other types of data would not be as good.

Here, the authors build a metapopulation model and use 2 different data sources and 2 different mobility models to feed it: CDRs from mobile phone data, commuting data from a census, and the gravity and radiation models, adjusted to match the census patterns. The outcome of the model is compared to weekly time series of number of cases at the level of departments. The authors first compare the origin destination matrices of the 4 mobility data sources, highlighting both similarities and differences.They then show that the model is able in all cases to reproduce the epidemic peak, and yields a high correlation between the model and measured incidences at the country level. 

When going into more details, by looking at the time series of the model's weekly incidence vs the observed weekly incidence in each department, they show that the best results are obtained with the CDR mobility data. Moreover, the performance of the CDR data is lower for departments where population and mobility fluxes are lower, while for the other data types performance is very heterogeneous.

Overall, the manuscript is well written, performs a timely and very relevant study, and certainly deserves publication. I have two major points, on two very different aspects:

-it seems that data is not made available because it stems from mobile phone data. While I understand that the raw CDR data cannot be shared because it is proprietary, in the end the authors here use only very aggregated data, namely averaged flows at the scale of departments. I do not understand how such data cannot be shared, and I think it is crucial than an effort is made in this direction and the origin-destination matrices are made publicly available.

-the gravity and radiation models are adjusted to the census data. However, as the authors note, the census data is from 2005. Therefore, it would seem more natural to adjust these models to the CDR data, to allow for a fairer comparison. Moreover, it would be then extra-interesting, if the gravity and/or radiation models were to perform well, as the resulting origin-destination matrices could then act as a substitute to data that cannot be shared.

Reviewer #2: The manuscript "Comparing sources of mobility for modeling the epidemic spread of Zika virus in Colombia" provides a comparison of different mobility metrics including CDR data to model the spread of Zika across departments in Colombia. The paper is well written and provides a useful comparison of the different mobility patterns and how these impact model estimates. Below are some comments to address: 

Major comments

1. Line 78: I see references for a few papers using mobility data for dengue among others – Are there other papers on ZIKA that include CDR mobility? Worth including if so 

2. Line 183: Could the authors expand on how the flows were rescaled by the population for the Census Network? 

3. Line 303: in each of the panels in Figure 4, there’s a noticeable outlier – did the authors evaluate the change in the correlation estimate with the removal of the outlier, or how much it might be impacting those trends? 

4. Line 322: How does the model’s estimated peak incidence compare to what was officially reported? 

5. Line 361: How does the model estimated total case compare to the observed? 

Line 452: Are there seroprevalence estimates for Colombia or particular departments that can be compared with the estimates derived from the model? 

Minor comments: 

1. Will the R code be made publicly available?

PLOS authors have the option to publish the peer review history of their article (what does this mean?). If published, this will include your full peer review and any attached files.

Reviewer #1: No

Reviewer #2: No

Figure Files:

Data Requirements:

Reproducibility:

References

---

## [Decision Letter · Decision Letter 1]

6 Jun 2022

Dear Dr. Perrotta,

We are pleased to inform you that your manuscript 'Comparing sources of mobility for modelling the epidemic spread of Zika virus in Colombia' has been provisionally accepted for publication in PLOS Neglected Tropical Diseases.

Best regards,

Kate Zinszer

Associate Editor

Bruce Lee

Deputy Editor

Reviewer's Responses to Questions

**Key Review Criteria Required for Acceptance?**

**Methods**

-Are the objectives of the study clearly articulated with a clear testable hypothesis stated?

-Is the study design appropriate to address the stated objectives?

-Is the population clearly described and appropriate for the hypothesis being tested?

-Is the sample size sufficient to ensure adequate power to address the hypothesis being tested?

-Were correct statistical analysis used to support conclusions?

-Are there concerns about ethical or regulatory requirements being met?

Reviewer #1: (No Response)

Reviewer #2: (No Response)

**Results**

-Does the analysis presented match the analysis plan?

-Are the results clearly and completely presented?

-Are the figures (Tables, Images) of sufficient quality for clarity?

Reviewer #1: (No Response)

Reviewer #2: (No Response)

**Conclusions**

-Are the conclusions supported by the data presented?

-Are the limitations of analysis clearly described?

-Do the authors discuss how these data can be helpful to advance our understanding of the topic under study?

-Is public health relevance addressed?

Reviewer #1: (No Response)

Reviewer #2: (No Response)

**Editorial and Data Presentation Modifications?**

Reviewer #1: (No Response)

Reviewer #2: (No Response)

**Summary and General Comments**

Reviewer #1: The authors have considered all comments by the reviewers and answered in a satisfactory way. I particularly thank them for including the additional network I had suggested.

I have only spotted one minor point: in the caption of figure 5 the color blue is cited twice, one should be "green" instead.

I recommend publication of this very nice paper.

Reviewer #2: (No Response)

PLOS authors have the option to publish the peer review history of their article (what does this mean?). If published, this will include your full peer review and any attached files.

Reviewer #1: No

Reviewer #2: No

---

## [Editor Report · Acceptance letter]

8 Jul 2022

Dear Dr. Perrotta,

We are delighted to inform you that your manuscript, "Comparing sources of mobility for modelling the epidemic spread of Zika virus in Colombia," has been formally accepted for publication in PLOS Neglected Tropical Diseases.

Best regards,

Shaden Kamhawi

co-Editor-in-Chief

Paul Brindley

co-Editor-in-Chief
